# Protein methyltransferase 7 deficiency in *Leishmania major* increases neutrophil associated pathology in murine model

**Juliana Alcoforado Diniz**[1], **Mariana M. Chaves**[2¤], **Slavica Vaselek**[3], **Rubens D. Miserani Magalhães**[1], **Rafael Ricci-Azevedo**[1], **Renan V. H. de Carvalho**[1], **Lucas B. Lorenzon**[1], **Tiago R. Ferreira**[2], **Dario Zamboni**[1], **Pegine B. Walrad**[4], **Petr Volf**[3], **David L. Sacks**[2], **Angela K. Cruz**[1]*

**1** Department of Cell and Molecular Biology, Ribeirão Preto Medical School, University of São Paulo, Ribeirão Preto, São Paulo, Brazil, **2** Laboratory of Parasitic Diseases, National Institute of Allergy and Infectious Diseases, National Institutes of Health, Bethesda, Maryland, United States, **3** Department of Parasitology, Charles University, Prague, Czech Republic, **4** Department of Biology, University of York, York, United Kingdom

¤ Current address: Department of Cell and Molecular Biology, University of São Paulo, Ribeirão Preto, São Paulo, Brazil
* akcruz@fmrp.usp.br

## Abstract

*Leishmania major* is the main causative agent of cutaneous leishmaniasis in the Old World. In *Leishmania* parasites, the lack of transcriptional control is mostly compensated by post-transcriptional mechanisms. Methylation of arginine is a conserved post-translational modification executed by Protein Arginine Methyltransferase (PRMTs). The genome from *L. major* encodes five PRMT homologs, including the cytosolic protein associated with several RNA-binding proteins, *LmjPRMT7*. It has been previously reported that *LmjPRMT7* could impact parasite infectivity. In addition, a more recent work has clearly shown the importance of *LmjPRMT7* in RNA-binding capacity and protein stability of methylation targets, demonstrating the role of this enzyme as an important epigenetic regulator of mRNA metabolism. In this study, we unveil the impact of PRMT7-mediated methylation on parasite development and virulence. Our data reveals that higher levels of *LmjPRMT7* can impair parasite pathogenicity, and that deletion of this enzyme rescues the pathogenic phenotype of an attenuated strain of *L. major*. Interestingly, lesion formation caused by *LmjPRMT7* knockout parasites is associated with an exacerbated inflammatory reaction in the tissue correlated with an excessive neutrophil recruitment. Moreover, the absence of *Lmj*PRMT7 also impairs parasite development within the sand fly vector *Phlebotomus duboscqi*. Finally, a transcriptome analysis shed light onto possible genes affected by depletion of this enzyme. Taken together, this study highlights how post-transcriptional regulation can affect different aspects of the parasite biology.

**Data Availability Statement:** All relevant data are within the manuscript and its Supporting information files.

**Funding:** This project was supported by São Paulo Research Foundation, FAPESP (https://fapesp.br/) MRC/FAPESP, 2015/13618-8 to AKC; Newton Fund Medical Research Council [MR/M02640X/1, MR/N017633/1] (https://mrc.ukri.org/), grants to PBW; Brazilian National Council for Scientific and Technological Development (https://www.gov.br/cnpq/pt-br), CNPq (305775/2013-8) to AKC. This study was supported in part by the Coordenação de Aperfeiçoamento de Pessoal de Nível Superior – Brasil (CAPES - https://www.gov.br/capes/pt-br) – Finance Code 001, AKC and DZ. The experiments on sand flies were supported by the European Commission, Horizon 2020 Infrastructure Infravec2 project (https://infravec2.eu) and ERD Funds, project CePaViP CZ.02.1.01/0.0/0.0/16_019/0000759) to PV and SV. The studies were funded in part by the intramural program of the National Institute of Allergy and Infectious Diseases (https://www.niaid.nih.gov/) to DLS. During the course of this work, JAD (grants 2016/14657-0 and 2018/02761-2), RRA (grant 2017/02998-0), RDMM (2019/18607-5) and LBL (grant 2016/00969-0) were supported by FAPESP (Fundação de Amparo a Pesquisa do Estado de São Paulo) fellowships. The funders had no role in study design, data collection and analysis, decision to publish, or preparation of the manuscript.

**Competing interests:** The authors have declared that no competing interests exist.

## Author summary

Understanding the genetics of *Leishmania*, a protozoan parasite causing leishmaniasis, is relevant for understanding fundamental questions on the pathogen's biology and its interaction with hosts. We explore mechanisms used by *Leishmania* to promptly adapt to different hosts investigating the control of gene expression occurring at the post-transcriptional level in the parasite. Methylation of arginine performed by Protein Arginine Methyltransferase (PRMTs), among other post-translational modifications, may alter the function and interactions of target proteins, some of them are RNA binding proteins, known regulators of gene expression. In this study, we unveil the impact of PRMT7 on parasite development and pathogenicity. In addition to a negative correlation between the levels of *Lmj*PRMT7 and parasite pathogenicity, we observed an impairment of the parasite development in the sand fly vector. Remarkably, despite a severe lesion development in mice, we observed no differences in parasite burden between infections with the pathogenic *Lmj*PRMT7 knockout parasite or the attenuated parental line. Instead, the severe pathology observed is associated with an exacerbated inflammatory response correlated with excessive neutrophil recruitment.

## Introduction

Leishmaniasis is a vector-borne disease caused by unicellular protozoans of the genus *Leishmania*, affecting around 12 million people worldwide [1]. Cutaneous leishmaniasis (CL) is the most common form of leishmaniasis, and *Leishmania major* is the main causative agent of CL in the Old World [2]. CL is characterized by the presence of skin ulcers that are produced by immunopathogenetic mechanisms [2]. The immune response to *L. major* infection in mice involves the development of polarized Th1 or Th2 responses associated with disease resistance or susceptibility, respectively [3]. In *Leishmania*-resistant C57BL/6 mice, a Th1-oriented immune response is associated with IFNγ and IL-12 production. In *Leishmania*-susceptible BALB/c mice, IL-4 production leads to the development of uncontrolled lesions and visceralization [3].

To complete its lifecycle *Leishmania* alternates between a mammalian host and an insect vector, Phlebotomine sand flies of the genera *Phlebotomus* (Old World) and *Lutzomyia* (New World) [4]. Upon uptake by a female Phlebotomine during a bloodmeal, infected mammalian macrophages lyse and release *Leishmania* amastigotes into the sand fly midgut. Here these parasites differentiate into progressive promastigote stages proliferating and migrating anteriorly. Upon colonization of the stomodeal valve these differentiate into the mammalian-infective metacyclic stage which are transmitted to vertebrate hosts during subsequent bloodmeals [5]

Genetic organization at the structural and functional levels contribute to the parasite's ability to rapidly adapt to dramatically different hostile environments. The organization of genes in long polycistronic transcription units and lack of canonical RNA Polymerase II promoters culminates with transcription of the whole genome at similar levels, transferring to the post-transcriptional level the control of gene expression [6,7]. The several layers of control after transcription start with the regulation of mRNAs stability, their subcellular localization, and rate of translation followed by post-translational modifications and protein stability [8]. It is well established that RNA Binding Proteins (RBPs) play a central role in several of these control strategies, such as modulation of RNA decay or initiation of translation rates. In addition, RBPs are modifiable by distinct post-translation modification processes (PTMs), which may affect their binding activities and specificities [9].

Remarkably, RBPs are well-known substrates of Protein Arginine Methyltransferases (PRMTs) [10]. PRMT enzymes catalyze the transfer of a methyl group from a donor to the guanidino-nitrogen atoms of target arginine residues [11]. These enzymes are broadly distributed across eukaryotes and can be classified as Type I, II and III [12]. Previous work from our lab characterized a *Leishmania major* PRMT7 homolog *in vivo*, providing evidence that pathogenicity in mice could be increased in parasites lacking *Lmj*PRMT7 and attenuated in parasites overexpressing this enzyme [13]. In a recent work Ferreira *et al* have validated *in vivo* and *in vitro* possible targets for PRMT7 methylation and presented *Leishmania* PRMTs as epigenetic regulators of downstream parasite virulence [14]. However, the impact of PRMT7-mediated methylation on *Leishmania* promastigote to amastigote development and parasite virulence is still to be investigated.

Herein, we explore the role of *Lmj*PRMT7 in parasite pathogenicity in mice and differentiation in the sand fly vector. Working with a laboratory-attenuated *L. major* strain, we find that it expresses higher levels of *Lmj*PRMT7 compared to pathogenic strains and that deletion of *LmjPRMT7* leads to the recovery of pathogenicity in this strain. Interestingly, despite marked differences in lesion development, the parasite loads in both the ears and draining lymph nodes of infected mice were similar in knockout and control lines. Our data suggest that increased pathogenicity in *PRMT7* knockout parasites is not related to the mouse genetic background and to T cell response profile. It was rather associated with an exacerbated inflammatory reaction sustained by differential neutrophil recruitment to the site of infection during the chronic phase of the disease. In addition to neutrophil response, *Lmj*PRMT7 levels also regulate parasite lifecycle progression within the sand fly vector. Altogether, the data reveal an intriguing role for PRMT7 in post transcriptional control of *Leishmania* pathogenicity.

## Methods

### Ethics statement

All mouse experimental procedures were performed in accordance with the Ethical Principles in Animal Research approved by the Local Ethical Animal Committee (CEUA) of FMRP-USP (protocol 163/2017) and by the NIAID Animal Care and Use Committee (no. LPD 68E). Use of animals in this research was strictly monitored for accordance with the Animal Welfare Act, the Public Health Service Policy, the U.S. Government Principles for the Utilization and Care of Vertebrate Animals Used in Testing, Research, and Training, as well as the National Institutes of Health Guide for the Care and Use of Laboratory Animals.

### Parasites and mice infection

*Leishmania major* strains CC1 (MHOM/IR/83/LT252), LV39 (MRHO/SU/59/P), Sd (MHOM/SN/74/SD), Fn (MHOM/IL/80/Friedlin) and Ryan (MHOM/IQ/07/WR2885) were cultured as promastigotes *in vitro* at 26 ˚C in complete Medium 199 (M199) supplemented with 20% heat-inactivated fetal calf serum (Gemini Bio-products), 100 U/ml penicillin, 100 μg/ml streptomycin, 2 mM L-glutamine, 40 mM Hepes, 0.1 mM adenine (in 50 mM Hepes), 5 mg/ml hemin (in 50% triethanolamine), 1 mg/ml 6-biotin, and 50 μg/ml of Geneticin (G418, Gibco, Woodland, CA, USA). *L. major* CC1 transfectants (wild type, *Δlmjprmt7* and 'addback' *Δlmjprmt7*, which contains ectopic copies of the wild type *LmjPRMT7*) were previously generated [13]. Parasites overexpressing mCherry were generated for *in vivo* uptake analysis by electroporation with the *Swa*I digest fragment from plasmid pLEXSY-mCherry-SAT2 and subsequent selection in complete M199 medium containing 200 μg/mL nourseothricin (Jenabioscence). Drug-resistant clones were obtained by serial dilution and mCherry expression was evaluated on a FACSCANTO II flow cytometer (BD Biosciences).

Infective stage metacyclic promastigotes were isolated from stationary cultures (7d) by density gradient centrifugation as described previously [15]. Mice were then inoculated with 100,000 metacyclic promastigotes in the ear by intradermal injection in a volume of 10 μl. Uninfected mice were used as the control group. Lesion development was monitored weekly by measuring the ear thickness with a direct-reading Vernier caliper (Thomas Scientific).

Female BALB/c mice, 6–8 weeks of age, were maintained in the National Institute of Allergy and Infectious Diseases (NIAID) animal care facility under specific pathogen-free conditions and female BALB/c mice and female C57BL/6 mice were also maintained in the University of São Paulo animal care facility.

## Sand fly infection and parasite recovery

The colony of *P. duboscqi* (originating in Senegal) was maintained in the insectary of the Department of Parasitology, Charles University, under standard conditions (26˚C on 50% sucrose, humidity 60–70% and 14h light/10h dark photoperiod) as previously described [16]. At 24 hours prior to the infection feeding, groups of approximately 150 sand fly females (5–7 days old) were separated and deprived of sucrose food. Promastigotes from log-phase cultures were resuspended in heat-inactivated rabbit blood at concentration of $10^6$ cells/mL and offered to sand flies through the chicken skin. Engorged females were separated, provided with sucrose, and maintained under standard conditions until the end of the experiment.

Dissections were performed before defecation (early stage of infection) on day 2 post blood meal (PBM) and after defecation (late stage of infection) on day 8 and 12 PBM. Abundance and localization of the flagellates in the sand fly gut were examined by light microscopy. Parasite loads were graded as light, moderate/medium, and heavy ($\leq$100, 100–1000, and $\geq$1000 parasites per gut, respectively) [17]. The number of parasites was also assessed by Burker chamber counting. Experimental feeding of *P. duboscqi* was repeated four times, 209 *P. duboscqi* females were dissected for localization/estimation analysis and an additional 125 sand flies were dissected for burker chamber counting.

## Quantitative real-time PCR

RNA from procyclic promastigote and metacyclic promastigote stages was extracted using TRIzol reagent (Invitrogen), chloroform extraction step and the 70% ethanol wash (both steps performed twice). The extracted RNA was treated with RNase-free DNase (Ambion) and DNase Inactivation Reagent (Ambion). cDNA was synthesized from 2 μg of RNA using the SuperScript Reverse Transcriptase (Invitrogen) according to manufacturer's instructions. An absolute quantification curve was performed for each oligonucleotide pair using serial dilutions of cDNA. Relative quantification was performed using the Power SYBR Green PCR Master Mix (Applied Biosystems) in 20 μL reactions. The PCR reactions were performed in an ABI 7500 thermocycler (Applied Biosystems). The following primers were used for quantification: RT_LmjF.06.0870_F (5′-CCATGAGCACCAGCAGCGAAA-3′), RT_LmjF.06.0870-v2_R (5′-GCGGCACGAAGATGTGCTGTT-3′), LLG6PD_F (5′-ACCGCATTGACCACTACCTC-3′), LLG6PD_R (5′-GATGTTGTTCGAGTTCCAC-3′), RNA45_F (CCTACCATGCCGTGTCCT TCTA) and RNA45_R (AACGACCCCTGCAGCAATAC).

## Immunoblot analysis

*Leishmania* cells were pelleted (7 min, 3 ˚C, 2000 x g), washed with 500 μl of ice-cold PBS containing 1X complete protease inhibitors (Roche) and resuspended in ~ 10 μl of extraction buffer (SDS 2%, 50 mM Tris-Cl pH 7,4, 1 mM PMSF, 1x complete) per 1 x $10^6$cells. The samples were boiled 10 min, quantified on Nanodrop One (Thermo Scientific), mixed with 0.2V

of sample buffer 6X (350 mM Tris-Cl pH 6.8, 30% glycerol, 10% SDS, 0.12 mg/mL bromophenol blue, 6% 2-mercaptoethanol) and boiled for 3 more minutes. ~ 40 μg of protein extract were loaded per well in polyacrylamide gels. The proteins were transferred to nitrocellulose membranes (GE Healthcare Life Sciences—REF 10600003), which were incubated with anti-monomethyl arginine antibody (Cell Signaling Technologies—REF 8015S), following manufacturer's instructions. The antigen-antibody interaction was detected using an ECL kit (GE-Healthcare), and chemiluminescence was visualized using ImageQuant LAS 4000 (GE-Healthcare).

## Processing of ear tissues and evaluation of parasite burden

Ear tissue was prepared as previously described [18]. In brief, the two sheets of infected ear dermis were separated, deposited in DMEM containing 0.2 mg/ml Liberase TL–purified enzyme blend (Roche Diagnostics Corp.), and incubated for 1.5 h at 37 ˚C. Digested tissue was processed in a tissue homogenizer for 3.5 min (Medimachine; Becton Dickinson) and filtered through a 70-μm cell strainer (Falcon Products). For preparation of the draining lymph node, in summary, the lymph nodes were extracted from the mice and macerated into a filter and washed with DMEM media. An aliquot of ear and lymph node homogenates were serially diluted in 96-well, flat-bottom microtiter plates containing 100 μl M199/S for parasite titrations. The number of viable parasites in each ear and draining lymph node was determined from the highest dilution at which promastigotes could be grown out after 7–10 days of incubation at 26 ˚C.

## Immunolabeling and flow cytometry analysis

For immunolabelling, cells processed as above were washed and labeled with Live/Dead fixable AQUA at a 1:500 dilution of the manufacturer suggested stock solution (Invitrogen) to exclude dead cells. Cells were incubated with anti-Fc III/II (CD16/32) receptor Ab (2.4G2), followed by surface staining with various combination of the following antibodies for 30 min at 4 ˚C in the dark: FITC anti-Ly6G, APC anti-CD206, BV421 anti-SiglecF, APC-Cy7 anti-Ly6C, PE-Cy7 anti-CD11b, PerCp-Cy5.5 anti-CD64. All antibodies were from eBiosciences, BD Biosciences, Biolegends or R&D systems. For intracellular staining, the cells were fixated and permeabilized (Fixation/Permeabilization Concentrate and Fixation/Permeabilization Diluent, Invitrogen) for 20 min at 4 ˚C. After, the cells were washed twice (Fixation/Permeabilization Buffer, Invitrogen) and added the intracellular antibody for 30 min at 4˚C. The Abs used were: FITC anti-γδTCR, FITC anti-NK1.1, APC anti-CD3, PE-Cy7 anti-CD2, APC-Cy7 anti-CD45.2, PerCp-Cy5.5 anti-CD11b. Data were collected using FACSDiva software on a FACS-CANTO II flow cytometer (BD Biosciences) and analyzed using FlowJo software (TreeS-tar). To determine the absolute number of cells, a portion of each sample was removed for counting with AccuCheck Counting Beads (Invitrogen).

## Neutrophil isolation, leishmania uptake and NET-DNA release measurement

To obtain bone marrow derived neutrophils (BMDNs) the femurs and tibia were isolated and flushed with PBS. Neutrophils were sorted using the Neutrophil Isolation Kit, mouse (Miltenyi Biotec). The cells were infected with *L. major* strains. Uninfected/untreated neutrophils were used as controls. After 3 h post-infection, the cultures were cytocentrifuged (50 × g, 5 min) and stained with HEMA3 Differential Stain Procedure (Thermo Fisher). The number of neutrophils with intracellular parasites was determined by count among 200 cells/condition, using an Olympus BX50 microscope.

For NET formation assay, neutrophils ($2 \times 10^5$ cells/well in HBSS) were incubated with different multiplicities of infection (MOIs) of *L. major* strains and PMA (50 nM), for positive control, in a 96-well plate for 3 h at 37 °C, in 5% $CO_2$. NET-DNA was quantified by adding SYTOX green nucleic acid stain (Invitrogen; 1 μM) to the supernatant of each incubation in a black 96-well plate and incubating at 37 °C for 10 min. NET-DNA fragments were quantified using a FLx800 Fluorescence Microplate Reader (BioTek Instruments, USA; excitation 485 nm, emission 528 nm) and the results have been expressed as fold to control (HBSS).

### RNA-seq *in silico* analysis

Reads were sequenced with exceptional quality above Q30 (Phred quality score) per base for both sample groups (promastigote metacyclic—META, and promastigote procyclic—PRO). Trimmomatic v. 0.36 [19] was used to remove any Illumina adaptor remaining sequences, and FastQC v. 0.11.5 analysis [20] and multiQC tool v. 1.3.dev0 [21] to calculate the quality of the reads. Bowtie2 v.2.2.6 [22] was used for alignment of reads (local alignment, one mismatch maximum admitted in the alignment seed, reference genome "TriTrypDB-41_LmajorFriedlin"). Other quality parameters were checked using QUALIMAP v.2.2.1 tool [23]. A high percent of reads was properly mapped over the reference genome of the parasite (97,95%) in accordance with the high general alignment of reads (99,24%) (S1 Table). FeatureCounts v. 1.5.3 [24], was used to count the number of reads mapped over each CDS coordinate (GFF file, TriTryDB, version 32). The genes were filtered using Counts Per Million (CPM) at least 1 in all samples. Limma R package was applied to quantile normalization (Trimmed mean of M-Values—TMM method) [25], log2 transformation and mean variance adjustment of count values (Voom) [26]. Similarity between the samples was evaluated by MDS plot. Detection of Differential Expressed Genes (DEG) of each contrast was proceeded using Generalized Linear Model (GLM) method of Limma [27]. p-Values were adjusted for multiple testing using the Benjamini-Hochberg (BH) method and adjusted p-value above 5% and Fold Change (FC) threshold of 1.5 were applied for selection of DEG (S2 Table). The DEG of each contrast (Wild type (WT)- *Δlmjprmt7* (Δ7), *Δlmjprmt7* [PRMT7] (AB), AB-WT) were clustered by the expression significance level in WT, Δ7 or AB based on Limma T-test (p = 5%—decideTest fuction) and F-test between groups (p = 5%).

All the combinations of expression levels for PRO and META samples are shown in S3 Table. DEG clustered in the combinations 6 (Δ7 down-regulated in relation to WT and AB with no significative difference between WT and AB) and 21 (Δ7 up-regulated in relation to WT and AB with no significative difference between WT and AB) were considered as interesting targets in PRO and META samples. The expression of these genes would be only altered by deletion of PRMT7 and it would be recovered in the *'addback'* strain (S1 Fig).

## Results

### Deletion of *LmjPRMT7* impairs parasite development in *Phlebotomus duboscqi* vector

Previous work from our group had shown that deletion of *Lmj*PRMT7 from an attenuated strain of *Leishmania major* correlates with increased pathogenicity [13]. To assess if *Lmj*PRMT7 could play a role in the ability of the parasite to differentiate to metacyclics we first analyzed *in vitro* the expression of SHERP, a metacyclic-specific marker [28]. According to our data the levels of SHERP were drastically decreased in *Δlmjprmt7* parasites in the metacyclic stage when compared to the wild type parasites (S2 Fig).

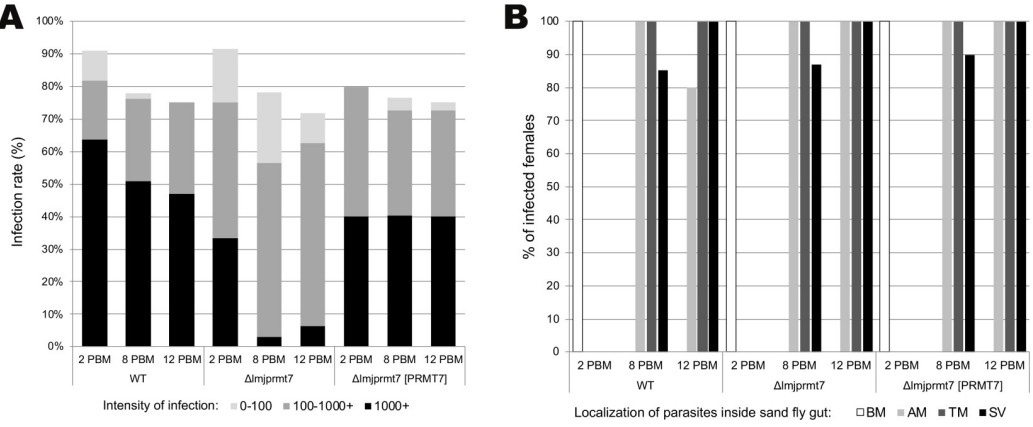

**Fig 1. Deletion of LmjPRMT7 results in a decrease in the number of parasites inside the *P. duboscqi* vector. (A)** Infection rates and parasite load in *P. duboscqi* on day 2, 8 and 12 post blood meal. Abbreviations: 0 no infection; 0–100 low infection; 100–1000 medium infection; 1000+ heavy infection. **(B)** Localization of parasites inside the *P. duboscqi* gut on day 2, 8 and 12 post blood meal. Abbreviations: SV—stomodeal valve, TM—thoracic midgut, AM—abdominal midgut, BM—within the bloodmeal. Data originate from 4 independent experiments; sand flies were infected by 10⁶ parasites per ml of bloodmeal.

We also investigated if *LmjPRMT7* could affect the growth and development of the parasite within the sand fly vector. We used *Phlebotomus duboscqi*, which is a known vector for *Leishmania major* [29]. Eight days post blood meal (PBM), similar/high infection rates were obtained in females infected with all the three strains (wild type: 75.92%, *Δlmjprmt7*: 76.92%, *Δlmjprmt7 [PRMT7]*: 78.94%) (Fig 1A). Localization of infection was also similar, the stomodeal valve (SV) of *P. duboscqi* was colonized in similar frequency among all tested strains—wild type: 85.29%, *Δlmjprmt7*: 89.74%, *Δlmjprmt7 [PRMT7]*: 86.84%, respectively (Fig 1B). However, heavy infections were more frequent in flies infected with the wild type (50.84%) and addback strain (40.25%) compared to the flies infected with *Δlmjprmt7* (2.89%), in which predominantly medium and low intensity infections were observed (Fig 1A).

On day 12 PBM, similar infection rates among all strains were observed (wild type: 77.72%, *Δlmjprmt7*: 73.91%, *Δlmjprmt7 [PRMT7]*: 77.78%,) (Fig 1A) and the stomodeal valve was colonized in all infected flies (Fig 1B). As on day 8 PBM, heavy parasite loads prevailed in flies infected with wild type and addback strains, while low and medium intensity infection dominated flies infected with *Δlmjprmt7* strain (Fig 1A).

## Levels of *Lmj*PRMT7 are negatively correlated with *Leishmania major* pathogenicity

The interaction of the attenuated *L. major* strain with the mammalian host was previously shown to be affected by the levels of *LmjPRMT7* [13]. Following up these findings we performed a quantitative PCR to evaluate the expression of *LmjPRMT7* in different strains of *Leishmania major*. For this assay we used the attenuated (LT252, clone CC1) strain of *L. major* in comparison with other known virulent strains of this parasite (Fig 2A). Interestingly, there is no difference in the levels of *LmjPRMT7* in all the strains tested in the sand fly-infective procyclic stage (early logarithmic phase—day 2). By contrast, in the mammalian-infective metacyclic stage (enriched metacyclics by Ficoll gradient—day 7) the levels of *PRMT7* transcript were higher in the attenuated strain in comparison with the virulent strains. Nevertheless, we observed no major differences in the monomethylation profile when comparing the attenuated with the evaluated pathogenic strains of *L. major* (S3 Fig).

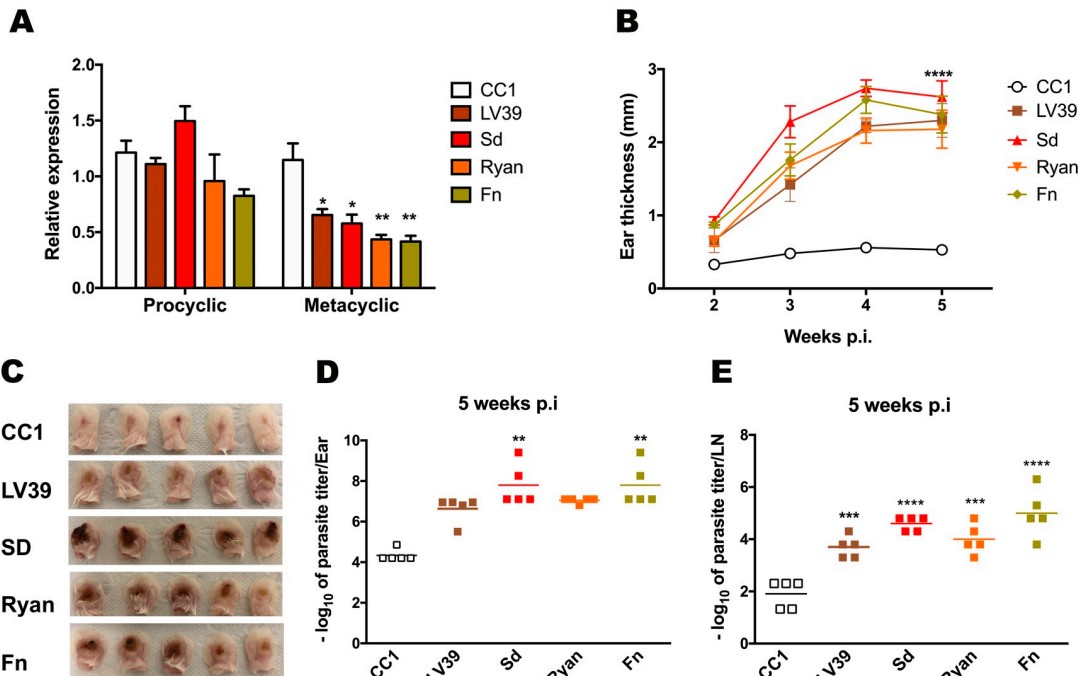

**Fig 2. Higher levels of PRMT7 enzyme are found in a nonpathogenic strain of *L. major*.** **(A)** Quantitative RT-PCR analysis of RNA extracted from promastigotes during the early-log (procyclic) culture phase and purified metacyclic. Levels of PRMT7 were comparable between the pathogenic *Leishmania major* strains: Ryan, LV39, Sd and Friedlin (Fn) in comparison to the nonpathogenic CC1 strain. The expression is relative to the expression of the G6PD and RNA45 genes. The results are shown as mean ± SEM. p < 0.05 (*) and p < 0.01 (**) were considered statistically significant. Statistical analysis was performed by Two-way ANOVA followed by Sidak post-test. **(B)** BALB/c mice (n = 5 per group) were injected with $10^5$ metacyclic promastigotes of the different strains of *L. major* and ear thicknesses were followed weekly. **(C)** Image of infected ears 5 weeks post infection and photos were taken by the author. Five weeks after infection, parasite titers were determined in the ear **(D)** and draining lymph node **(E)**. The results are shown as mean ± SEM. For the parasite burden, statistical analysis was conducted comparing virulent strains with the *Lmj*CC1 attenuated line. Statistical analysis was performed by unpaired, two-tailed student's t-test. *p<0.05.

To confirm that *L. major* CC1 is less pathogenic than the other tested strains we monitored the thickness of BALB/c ear lesions *in vivo* for 5 weeks post infection (Fig 2B). As expected, infection with *L. major* CC1 strain caused minor lesions during the period analyzed. By contrast, each of the other strains produced uncontrolled, ulcerative lesions (Fig 2C). To verify possible differences in parasites proliferation and survival in the site of infection in the different strains after 5 weeks of infection, we quantified the parasite burdens in the infected ears (Fig 2D) and the draining lymph nodes (Fig 2E). As expected, there was a positive correlation between lesion formation and parasite load, with the number of parasites significantly lower in the attenuated *L. major* CC1 strain.

## Deletion of *LmjPRMT7* from the attenuated strain of *L. major* rescues pathogenicity

To determine if the elevated expression of *LmjPRMT7* is causally associated with the attenuated phenotype of the *L. major* CC1 strain, we examined whether CC1 lacking *LmjPRMT7* could rescue a pathogenic phenotype. Indeed, an analysis of the progression of BALB/c mice lesions infected with *L. major* CC1 transfectants (wild type, *Δlmjprmt7* and 'addback' *Δlmjprmt7*, which contains ectopic copies of the wild type *LmjPRMT7*) for 7 weeks showed that the *LmjPRMT7* knockout strain led to a significant increase in the lesion size (Fig 3A). In

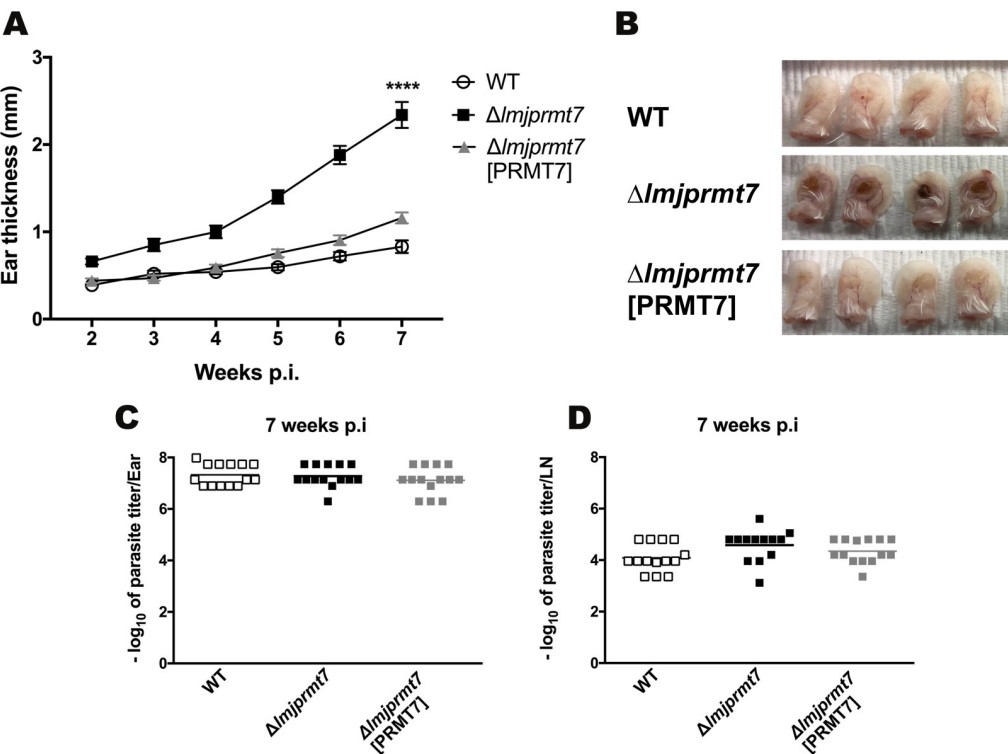

**Fig 3. Deletion of *Lmj*PRMT7 in a nonpathogenic strain leads to lesion formation not affecting parasite load. (A)** BALB/c female mice (n = 5 per group) were injected intradermally in both ears with 100,000 metacyclic promastigotes and ear thicknesses was followed for 7 weeks. **(B)** Photographs of infected ears on the 7th week post infection, taken by the author. Results shown are the mean ± SEM of twenty ears, five mice/group, from two independent experiments, $^{**}p < 0.01$, $^{****}p < 0.0001$, comparing infection with wild type and *Δlmjprmt7*, using 2way ANOVA, with Tukey's multiple comparison test. **(C-D)** Parasite burden determined by limiting dilution analysis (LDA) in the site of infection (ear, left) and draining lymph node (LN, right) after 7 weeks post-infection. Results shown are the mean ± SEM of fourteen ears from two independent experiments. Unpaired, two-tailed student's t-test.

addition, the addback strain complemented perfectly the non-pathogenic profile of the wild type *L. major* CC1 strain (Fig 3A and 3B).

 Remarkably, despite significant difference in lesion size, comparable parasite burdens were observed for transfectant and wild type parasites in both the ears (Fig 3C) and the draining lymph nodes (Fig 3D) 7 weeks post infection. Importantly, we find that removal of *LmjPRMT7* from the attenuated strain recovers parasite pathogenicity, but this pathology no longer correlates with parasite burden.

## Neutrophils are the main myeloid cell subset associated to the pathogenicity

Since there was no difference in the parasite load between the genetically modified strains of *L. major* CC1, we characterized the inflammatory and immune cells present in the lesions that could account for the different pathologic outcomes. The ears of the infected BALB/c mice were extracted and processed 7 weeks post infection, as previously described [18]. The gates used to define the myeloid cell subsets in the control group are shown in Fig 4A. Comparing the different immune cell subsets, neutrophils were preferentially recruited to the site of infection with *Δlmjprmt7* (Fig 4B). The other resident and recruited cell populations did not differ in their total numbers per ear comparing the wild type and transfected *L. major* strains.

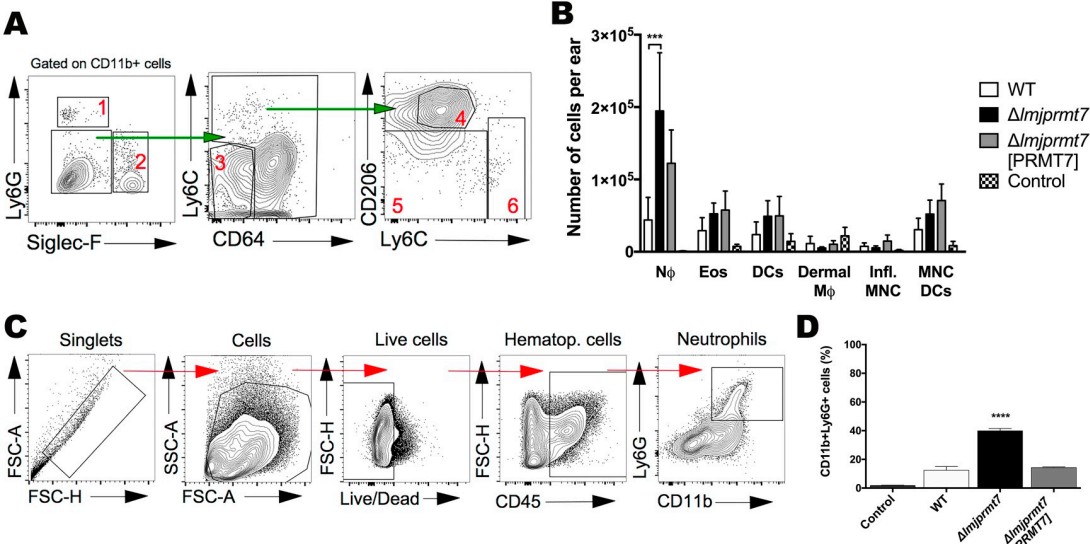

**Fig 4. Subset of myeloid cells recruited to the site of infection after 7 weeks post-infection.** BALB/c mice were infected in the ear dermis with 100,000 *Lmj* CC1 metacyclic promastigotes from the wild type, Δ*lmjprmt7* and Δ*lmjprmt7* [PRMT7] strains. Ear tissues were processed at 7 weeks post-infection to study the cell recruitment by flow cytometry. **(A)** Representative dot plots of ear-derived dermal cells. Subpopulations of myeloid (CD11b+) cells are defined as 1. Neutrophils, 2. Eosinophils, 3. Conventional dendritic cells, 4. Dermal macrophages, 5. Monocyte derived dendritic cells, 6. Inflammatory monocytes. **(B)** The number of cells in each myeloid cell subset after 7 weeks of infection (NΦ: neutrophils; Eos: eosinophils; DCs: dendritic cells; Dermal MΦ: dermal macrophages; Infl. MNC: inflammatory monocytes; MNC DCs: Monocyte derived dendritic cells). All data are shown as the mean ± SEM of twenty samples (five mice, per group) from two independent experiments ***p<0.001, using ordinary one-way ANOVA, with Dunn's multiple comparison test. **(C)** The representative contour plot of another independent experiment and **(D)** the percentages of neutrophils (CD11b+Ly6G+ cells) after 7 weeks of infection.

In a repeat experiment in BALB/c mice (Fig 4C and 4D), we confirmed that neutrophils were preferentially recruited to the site of infection in the Δ*lmjprmt7* group compared to the wild type and addback infected groups after 7 weeks post infection. This led us to postulate that the accumulation of neutrophils might be responsible for lesion formation in the ears of mice infected with the mutant strain. We also investigated the neutrophils' recruitment in the early infection (24h and 72h p.i.). We observed a transient initial wave of neutrophil recruitment with a slight increase in neutrophil numbers in the knockout line after 24h and no difference among the strains after 72h. The percentage of neutrophils in the infected ears was low during these initial time points post infection (S4 Fig).

## Lesion formation by Δ*lmjprmt7* is similar in a susceptible and resistant mice genetic background

To further refine this immune contribution to PRMT7-dependent pathology, we analyzed the effect that the different transfectants of *L. major* CC1 strain would have on a C57BL/6 mice "resistant" background. Interestingly, we verified that in the wild type and PRMT7-complemented strains there was no lesion formation, while in the knockout mutant there was a clear lesion formed in the ear after 7 weeks post infection (Fig 5A). The severe pathogenic phenotype produced by the Δ*lmjprmt7* strain in susceptible BALB/c mice is shown for comparison (Fig 5B). After quantifying the parasite loads in the infected ears, we observed not only a similar number of parasites among the different transfectants but also in the comparison between C57BL/6 and BALB/c mice (Fig 5C and 5D). Of note, despite the different genetic background

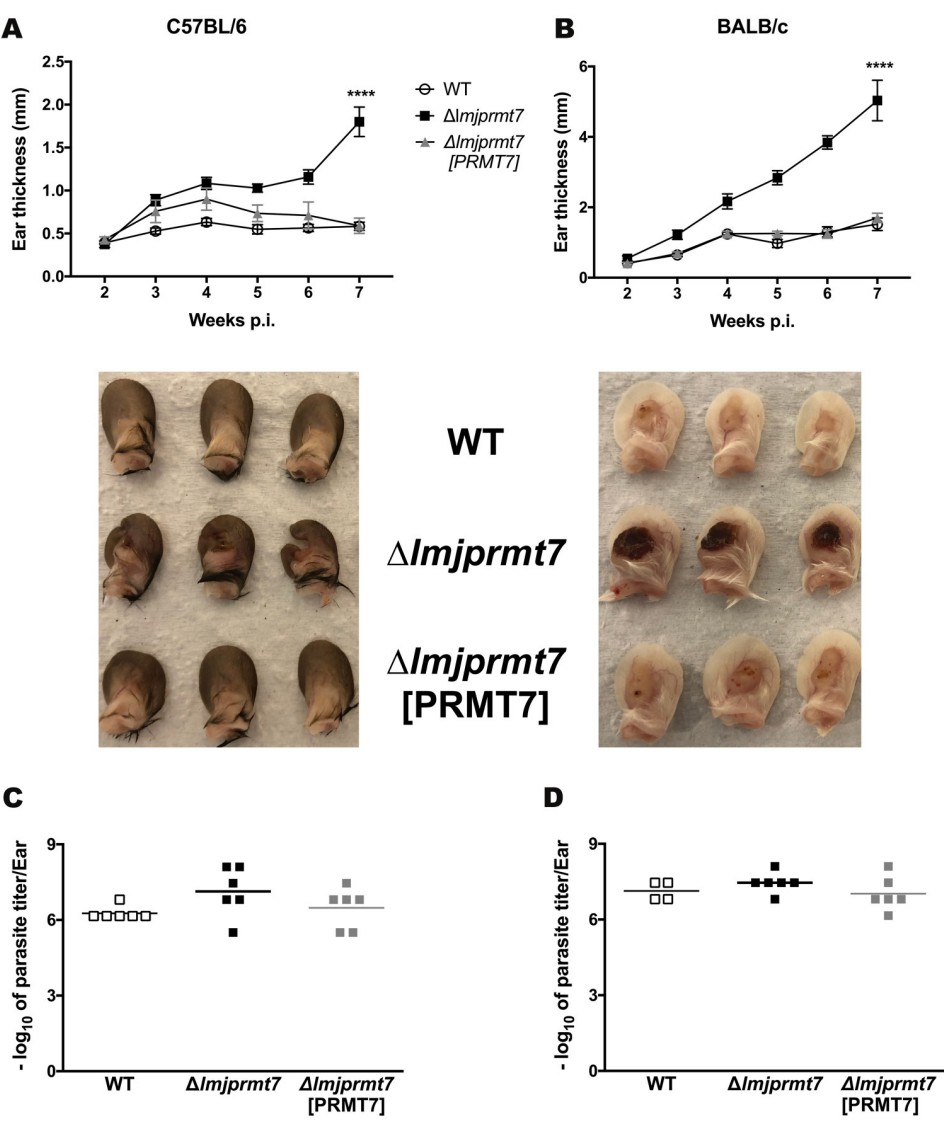

**Fig 5. Infection of resistant and susceptible mice strains display similar lesion outcome. (A)** C57BL/6 and **(B)** BALB/c mice were injected in the ear with $10^5$ metacyclic promastigotes of Lmj CC1 WT and transfectants (n = 6 mice per group), and the ear thicknesses were followed for 7 weeks. Images of infected ears are shown next to each graph as follow: 1 WT, 2 Δ*lmjPRMT7* and 3 Δ*lmjPRMT7* [PRMT7] and pictures were taken by the author. **(C-D)** Parasite quantification at 7 weeks post-infection. The results are shown as mean ± SEM. Statistical analysis was performed by unpaired, two-tailed student's t-test.

and immune response from the different mice model used (Th1 or Th2 polarized response), the infection outcome was indistinguishable.

### *In vitro* assay indicates higher uptake of *Δlmjprmt7* parasites by neutrophils

To examine specific leukocyte response to *Lmj*PRMT7 levels, we evaluated the behavior of these transfectants *in vitro*. First, we investigated the role of *LmjPRMT7* in the parasite uptake by neutrophils *in vitro*. Neutrophils were isolated from BALB/c mice bone marrow and the cells were incubated with serum opsonized, metacyclic promastigotes of the different

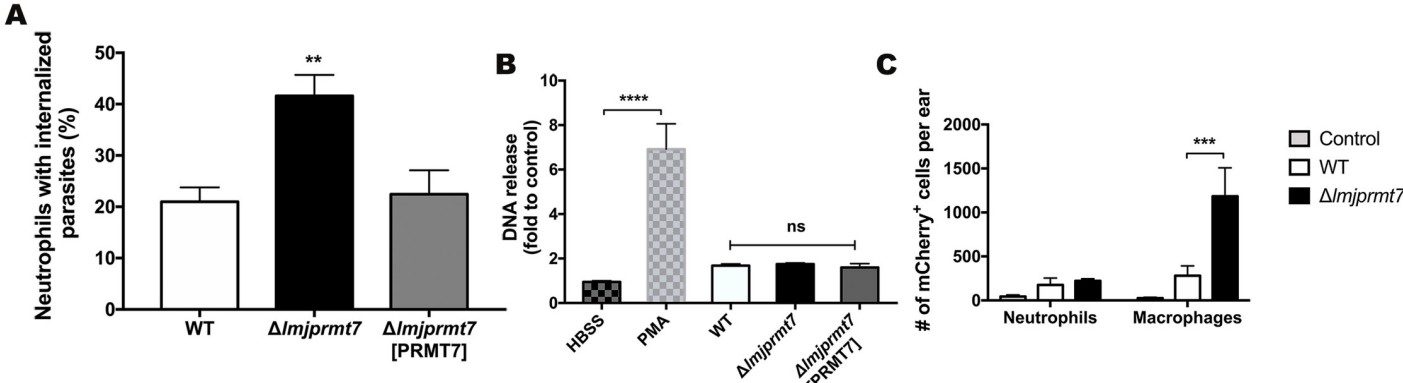

**Fig 6. Neutrophils display increased uptake of parasites lacking PRMT7 in *in vitro* assays. (A)** To evaluate the uptake of parasites by neutrophils, 10,000 BMDN were incubated for 3 hours with metacyclic promastigotes (MOI 1). The cells were centrifuged on slides and stained for evaluation by light microscopy. Data are expressed as the mean of neutrophils with intracellular parasites ± SEM. ** p<0.01. Statistical analysis was performed by ordinary one-way ANOVA followed by Tukey's multiple comparison test. These results are a pool of 3 independent experiments. **(B)** NET formation was assessed after incubating 10,000 BMDNs with modified strains of *L. major* CC1, PMA (50nM), or medium (untreated) at MOI of 1 for 3 hours. DNA release after the treatment/infection was quantified in the cell supernatants by fluorescence detection (Ex./Em. 480/520 nm) after reaction with SYTOX green. Data are expressed as a mean of fluorescence intensity ± SEM; **** p<0.0001. Statistical analysis was performed by two-way ANOVA followed by Tukey's multiple comparison test. DNA quantitation assays were performed in duplicate and, the data shown are the pool of two independent experiments. **(C)** To evaluate the uptake *in vivo* by flow cytometry, 4x10^5 *Lmj* CC1 metacyclic promastigotes from the wild type and *Δlmjprmt7* fluorescently tagged with mCherry were used to infect BALB/c mice. Ear tissues were processed at 4-hours post-infection to study the number of macrophages and neutrophils infected with the different fluorescent strains. Data are shown as the mean ± SEM of four samples per group. Representative data from two independent experiments. *** p<0.001, using two-way ANOVA followed by Tukey's multiple comparison test.

*Leishmania* strains. Interestingly, the uptake of the knockout mutant parasites by neutrophils was higher than that of wild type or *LmjPRMT7* complemented parasites (Fig 6A). We considered that this more efficient uptake might be associated with greater release of Neutrophils Extracellular Traps (NETs). While we detected no difference in DNA release by neutrophils infected with the different strains in this *in vitro* assay (Fig 6B), we cannot rule out possible *in vivo* stimulation not inducible in the given conditions.

We also tested the *in vivo* uptake capability of the wild type and knockout strains after 4 hours post-infection during the initial transient wave of neutrophil recruitment to the skin. The *in vivo* uptake of fluorescent strains by macrophages is consistent with our previous published data [13] in which the phagocytosis is increased in the knockout strain (Fig 6C). By contrast, the neutrophil uptake *in vivo* did not mimic the phenotype seen for the *in vitro* assay (Fig 6C). Taken together, the *in vitro* assay revealed that the knockout mutant strain increases the uptake by neutrophils in a mechanism that apparently does not involve host mediators to either recruit or activate neutrophils, suggesting that other factors not related to the host might be involved.

### Few differentially expressed transcripts are detected in the *LmjPRMT7* compared transfectants

To investigate possible parasite factors explaining the pathology induced by the *Lmj*PRMT7 knockout parasites, we searched for those genes differentially expressed in *Lmj*PRMT7 knockout parasites compared to the controls in procyclic and metacyclic promastigotes. Differentially expressed genes (DEG) were clustered in 27 groups obtained by comparison of each strain with one another (WT-Δ7, Δ7-AB and AB-WT) and 3 possible relative expression outcomes (up-regulated, down-regulated and no difference). This approach generated a list of genes that were DE only in *Δlmjprmt7* with no differences detectable between control lines (S2 Table). DEGs clustered by their presence in each comparison are shown in Venn diagrams

(Fig 7A). Four out of the seven DEGs from procyclic samples code for Histone H4 (LmjF.21.0015, LmjF.35.0015, LmjF.36.0020, LmjF.02.0020); all of which are downregulated in *Δlmjprmt7* versus wild type and addback strains. The other three genes are *galactose oxidase* (LmjF.28.1590), *phosphonopyruvate decarboxylase* (LmjF.28.1580) and *octanoyltransferase* (LmjF.36.3080). This last gene is the only one that was found upregulated in *Δlmjprmt7* compared to wild type and addback strains. Genes coding for *phosphonopyruvate decarboxylase* (LmjF.28.1580) and a *hydrolase, alpha/beta fold family* (LmjF.28.1570) were downregulated in *Δlmjprmt7* metacyclics (Fig 7B). Interestingly, *galactose oxidase* (LmjF.28.1590) and *phosphonopyruvate decarboxylase* (LmjF.28.1580) are present in the same region on chromosome 28 in a convergent SSR (Fig 7C). PRMT7 targets numerous RNA binding proteins impacting protein stability and regulatory function. This and PRMT intra-regulation offer some explanation to the complexity of these outcomes.

## Discussion

Herein, we explored the role of the *L. major* Protein Arginine Methyltransferase 7 (*Lmj*PRMT7) in parasite pathogenicity in a mouse model of CL and in parasite development in the sand fly vector. We demonstrated that there is a negative correlation between levels of PRMT7 and pathogenicity. Our results showed that this inflammatory response is observed in both *L. major*-susceptible and -resistant mouse strains. Of interest, PRMT7-dependent pathology does not correlate with parasite burden but instead correlates with a sustainable recruitment of neutrophils to the site of infection during the chronic phase of the disease. In addition, our sand fly data revealed a lower number of promastigotes in the sand fly vector when infected with the strain lacking *Lmj*PRMT7.

Ferreira *et al* have shown that levels of *Lmj*PRMT7 transcript in one pathogenic strain of *L. major* (LV39) were higher in procyclic than in stationary phase promastigotes [13]. We expanded this investigation by evaluating the levels of *Lmj*PRMT7 transcript during early-log-phase and metacyclic promastigotes of four different pathogenic strains of *L. major* compared with the attenuated *L. major* CC1 strain (Fig 2A). Interestingly, we realized that in metacyclics from the non-pathogenic strain the levels of *Lmj*PRMT7 were much higher in comparison with the others.

Regarding a possible role of *LmjPRMT7* in parasite differentiation we initially verified that the levels of SHERP were diminished in the metacyclic life stage of *Δlmjprmt7* parasites (S2 Fig). It is known that the genetic locus encoding *Leishmania*-specific proteins, HASPs and SHERP, is essential for metacyclogenesis of *L. major* in *Phlebotomus papatasi* [28]. To better analyze if *LmjPRMT7* had a role in metacyclogenesis we tested the differentiation within another natural vector for *L. major*, *Phlebotomus duboscqi* [29]. Similarly to recent reports [30,31] we analyzed the infection rate in female sand flies after 8- and 12-days post blood meal. Our data suggest that indeed depletion of *LmjPRMT7* impacts parasite load within *P. duboscqi* (Fig 1A). However, it does not affect the ability of these parasites to migrate to the stomodeal valve (Fig 1B). Since parasites attached to the stomodeal valve cause damage to the chitin lining and epithelial cells of the valve, this facilitates the reflux of parasites to the midgut and it is a prerequisite for effective transmission to the vertebrate host [5]. Another important prerequisite for infection is the differentiation to metacyclic stages, which is highly infective for the mammalian host. The tentative quantification of metacyclics via RT-qPCR using SHERP as a marker of the stage failed because there were no detectable levels of *Leishmania* RNA in the sand fly gut. Of note, a low percentage of metacyclics in *L. major* attenuated strains has been previously reported [32].

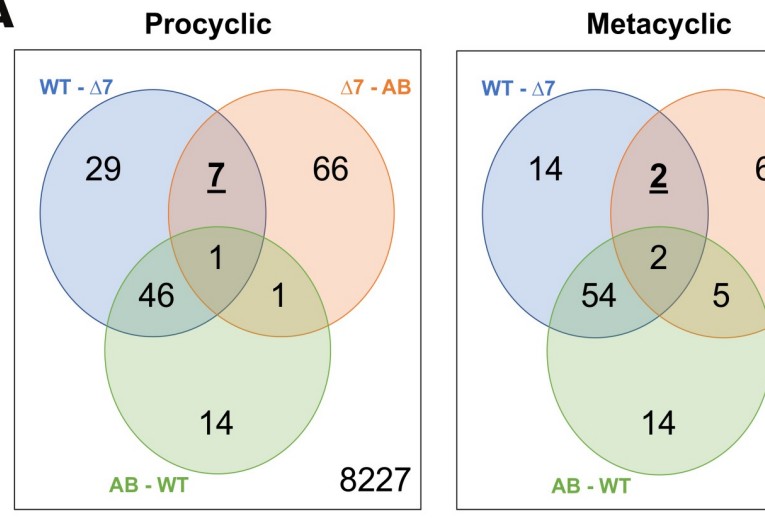

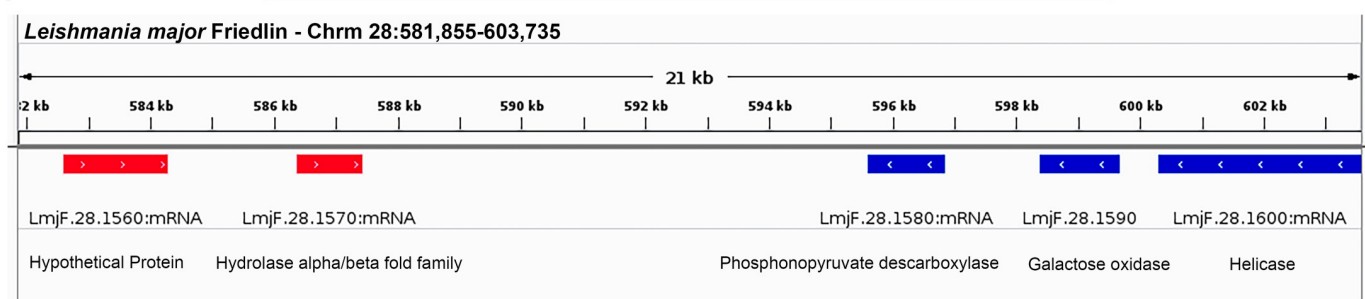

**Fig 7. Analysis of transcripts differentially expressed in the *LmjPRMT7* compared transfectants. (A)** Venn diagram of the three contrasts obtained after differentially expressed analysis of transcriptomes from *L. major* parasite samples (WT, Δ*lmjPRMT7* and Δ*lmjPRMT7* [PRMT7]). In bold the DEG only differentially expressed in Δ7 compared to WT and AB but not between the last two groups (FDR 5%, FC > 1.5). On the right bottom corner, the number of genes with no expression difference found between the groups. **(B)** Description of genes showed in bold in the Venn Diagram. **(C)** The convergent switch strand region on chromosome 28 depicting the location of three of the DEG identified, LmjF.28.1580, LmjF.28.1590 and LmjF.28.1570.

The impact of *LmjPRMT7* levels in leishmaniasis pathology was originally shown in our previous work [13]. A more recent study by Ferreira and coworkers demonstrates *Lmj*PRMT7 activity impacts protein target stability and function in a context-dependent manner [14]. A more robust study underlying how this modifying enzyme regulates host immune response was needed. Instead of the mouse footpad model used previously, we used the bespoke model for CL that more closely resembles the natural infection, involving intradermal infection in the ear dermis with a fraction enriched of metacyclic promastigotes [18]. In BALB/c mice that develop a strong Th2 response following infection with *L. major* strains, the outcome was the same as previously reported, with *Lmj*PRMT7 deletion rescuing the pathogenic phenotype of an attenuated strain (Fig 3A and 3B). While increased lesion size is typically associated with increased parasite loads [33–37], the reconstitution of lesion growth produced by the deletion of *LmjPRMT7* was not due to enhanced parasite survival or replication as the parasite burdens were similar in all strains examined (Fig 3C and 3D). Notably, this PRMT7-dependent pathology was not constrained in conventionally *Leishmania*-resistant, C57BL/6 mice, with the *Lmj*PRMT7 knockout parasites resulting in non-healing lesions similar to susceptible mice (Fig 5A). Even in parasite resistant strains, the parasite burden at the site of infection was comparable among all strains tested (Fig 5C). Taken together, these results suggest *PRMT7* knockout pathology is not associated with the mouse genetic background, favoring a specific T helper response profile.

According to our data, neutrophils were the main inflammatory cells present in the ear after 7 weeks of infection with the knockout mutant (Fig 4B). Furthermore, the number of neutrophils recruited to the site of infection were higher when ears were infected with parasites lacking *LmjPRMT7* (Fig 4D). Polymorphonuclear neutrophil granulocytes are known to be the first leukocytes to migrate to the site of infection and encounter the parasites. *Leishmania* have been demonstrated to use neutrophils as "Trojan horses" to silently enter the macrophage without immune response activation [38]. Similarly, the early anti-*Leishmania* response can also be inhibited by the capture of parasitized neutrophils in the skin by dermal dendritic cells [39]. Although, several elegant studies in different species of *Leishmania* have proven the rapid recruitment of neutrophils to the site of promastigote infection [40,41], the contribution of these cells to amastigote clearance, maintenance of inflammatory response and tissue damage is poorly understood. It has previously been suggested that augmented neutrophil proteases and inflammatory cytokines during the secondary wave of neutrophils (19 days pi) could contribute to skin inflammation, ulceration and necrosis in chronic lesions of *Leishmania Viannia* species [42]. In addition, it is well accepted that neutrophil accumulation in BALB/c mice contributes to susceptibility, and the increased levels of IL-17 are responsible for neutrophil migration and accumulation in leishmaniasis lesions [43]. On contrary, in *L. amazonensis* neutrophils may limit detrimental tissue damage during chronic disease [44]. Mounting evidence suggests that neutrophils play a complex and active role in parasite recognition and the anti-parasite response in both acute and chronic leishmaniasis [44]. Corroboratively, the *in vitro* data demonstrate that the *LmjPRMT7* knockout mutant increases the uptake by neutrophils (Fig 6A). In summary, our hypothesis is that the sustainable recruitment of neutrophils to the site of infection is responsible for an exacerbated inflammatory lesion formation in the ear of mice infected with the knockout mutant.

The lack of evidence of host immune response elements supporting the neutrophil recruitment and activation led us to search for parasite factors driving the different observed outcomes. Epigenetic control by PRMT7-dependent monomethylation of RBPs, the primary gene regulators in this system, can mediate parasite virulence months after PRMT7 expression [13,14]. In this context, we investigated which genes were differentially expressed in the absence of *LmjPRMT7* compared to the wild type and addback strains (Fig 7). Remarkably,

four Histone H4 genes were found downregulated in the parasites lacking *LmjPRMT7*. This data is in accordance with studies that have proven the essentiality of PRMTs methylation of Histone H4 *in vivo* [10,45]. Our collaborators have recently shown that *Lmj*PRMT7 can modify Histone H4 *in vitro*, however no histone MMA peptides were identified in a PRMT7 methyl-SILAC proteomics screen [14]. Of interest, multiple *Histone H4* genes were found downregulated in the absence of *LmjPRMT7* in the procyclic life stage. Another gene found differentially expressed, in this case upregulated in the knockout mutant, was the *octanoyl-transferase* (LmjF.36.3080). This gene, involved in lipoic acid biosynthesis in *L. major* [46], called *lipB*, has been found in different bacterial strains [47–49]. Interestingly, in *L. major* the overexpression of LIPB has been shown to impact metacyclogenesis [46] similarly to what we have seen in the *Δlmjprmt7* strain (S2 Fig). Another gene found downregulated in the *LmjPRMT7* knockout mutant, in both the procyclic and metacyclic stages, was the *phosphono-pyruvate decarboxylase* (LmjF.28.1580). The phosphonopyruvate decarboxylase (PnPDC) in *Bacteroides fragilis*, plays a key role in the formation of 2-aminoethylphosphonate, a component of the cell wall of this bacterium [50]. As such, PnPDC is a possible target for therapeutic intervention in this, and other phosphonate producing organisms [51]. In *Leishmania* spp, the key role of this enzyme is still unknown. However, a comparison of metabolic pathways encoded by the genomes of *T. brucei*, *T. cruzi*, and *L. major* revealed that *phosphonopyruvate decarboxylase* is part of the GPI anchor biosynthesis and its acquisition from bacteria occurred by horizontal gene transfer [52]. Finally, a gene found downregulated in the knockout mutant but only in the metacyclic stage was a *hydrolase, alpha/beta fold family* (LmjF.28.1570). Recently, this transcript was identified as upregulated in another RNA sequencing analysis looking for changes in the transcriptome of *L. major* promastigotes after a moderate heat shock [53]. Curiously, the 3 genes found downregulated in the knockout mutant sit in the same region in the genome of *L. major* facing each other in a convergent strand switch region (Fig 7C), suggesting a possible epigenetic modification of this genomic region on chromosome 28. Though preliminary, the RNA sequencing data demonstrated interesting targets to be further explored, particularly those functionally linked to metabolic regulation of membrane mediators associated with immune cells and/or attachment to the sand fly midgut.

In conclusion, our data supports that the attenuation of the pathogenesis of *L. major* CC1 strain is linked to the overexpression of a protein arginine-methyltransferase. This is corroborated by the strain pathogenicity being rescued by deletion of *LmjPRMT7*. Of interest, our data suggest a link between *Leishmania*-dependent lesion formation and excessive neutrophil-mediated inflammation and recruitment of these leukocytes up to 7 weeks p.i. This data indicates that perhaps the deletion of this enzyme might be promoting changes in the surface of the parasite which activate and/or recruit surplus neutrophils. Given the apparent similarity of arginine monomethylation profile among the *L. major* wild type strains, we cannot exclude that a putative PRMT7 moonlighting activity might be responsible for the KO phenotype. Our RNA sequencing analysis is merely the tip of the iceberg representing the first steps to interpret how *LmjPRMT7* controls *L. major* parasite virulence.

## Supporting information

**S1 Fig. Representation of one possible scenario of each interest group (S2 Table, rows 6 and 21).** This scenario depicts only those genes differentially expressed in *Δlmjprmt7* (Δ7) samples compared with wild type (WT) and *Δlmjprmt7* [PRMT7] (AB). Genes expressed at different levels in WT and AB samples were omitted. Down-regulated genes are shown on the left panel and upregulated on the right.
(TIFF)

**S2 Fig. The stage-specific gene SHERP is differentially expressed during development in *L. major*.** Quantitative RT-PCR analysis of RNA extracted from promastigotes during the early-log (procyclic) and purified metacyclic phases. Comparing *L. major* CC1 WT and *Δlmjprmt7* strains. The expression is relative to the expression of the G6PD and RNA45 genes. Statistical analysis was performed by Two-way ANOVA followed by Sidak post-test.
(TIF)

**S3 Fig. Arginine methylation profile of different strains of *L. major*.** Arginine monomethy-laded levels in procyclic (left panel) and enriched metacyclic (right panel) promastigotes. The pathogenic *Leishmania major* strains: Ryan, LV39, Sd and Friedlin (Fn) and the nonpathogenic CC1 strain were incubated with the anti-monomethylarginine antibody (α-MME) overnight at 4˚C. The lower panels show loading control with an antibody against the Elongation factor 1-α, as indicated (α-EF1α).
(TIFF)

**S4 Fig. Neutrophil recruitment is not increased during early infection.** BALB/c mice were infected in the ear dermis with 100,000 *Lmj* CC1 metacyclic promastigotes from the wild type, *Δlmjprmt7* and *Δlmjprmt7* [PRMT7] strains. Ear tissues were processed at 24- and 72-hours post-infection to study the cell recruitment by flow cytometry. All data are shown as the mean ± SEM of four samples per group. Representative data from two independent experiments. ** $p < 0.01$ and **** $p < 0.0001$, using two-way ANOVA followed by Tukey's multiple comparison test.
(TIFF)

**S1 Table. Mappability information for sequenced reads from each sample over reference genome ("TriTrypDB-41_LmajorFriedlin").**
(XLSX)

**S2 Table. Differential Expressed Genes (DEG) for each contrast (Wild type (WT)-*Δlmjprmt7* (Δ7), Δ7- *Δlmjprmt7*[PRMT7] (AB), AB-WT).** The table sheets show the genes found differentially expressed in each contrast considering an adjusted p-value of 0.05 or in addition to the adjusted p-value, a fold change value of at least 1.5. The columns show the gene ID (TritrypDB IDs), description and length ("ID", "Description", "Length" respectively), the estimate of the log2-fold-change for each contras ("log2FC"), average log2-expression ("AveExpr"), moderated t-statistic ("t"), raw p-value ("P.Value") and adjusted p-value or q-value ("adj.P.Val").
(XLSX)

**S3 Table. Clusterization of DEG by expression significance in each analyzed contrast.** The numbers: **1** means "up-regulated", **-1** "down-regulated", **0** "no significative expression difference" of transcripts level in a given contrast. The groups of PRO contrasts are shown on the left and the groups of META contrasts on the right. The groups of interest are highlight in green and red groups are those only possible from the mathematical standpoint.
(XLSX)

## Acknowledgments

We thank Tania P. A. Defina for all the help with the quantitative PCR analysis. We are also grateful to Djalma S Lima Jr for help with animal handling and *in vitro* assays. We thank Viviane Trombela for her technical assistance.

## Author Contributions

**Conceptualization:** Juliana Alcoforado Diniz, Dario Zamboni, Pegine B. Walrad, Petr Volf, David L. Sacks, Angela K. Cruz.

**Formal analysis:** Juliana Alcoforado Diniz, Mariana M. Chaves, Slavica Vaselek, Rubens D. Miserani Magalhães, Rafael Ricci-Azevedo, Renan V. H. de Carvalho, Tiago R. Ferreira.

**Funding acquisition:** Petr Volf, Angela K. Cruz.

**Investigation:** Juliana Alcoforado Diniz, Mariana M. Chaves, Slavica Vaselek, Rubens D. Miserani Magalhães, Rafael Ricci-Azevedo, Renan V. H. de Carvalho, Lucas B. Lorenzon, Tiago R. Ferreira.

**Methodology:** Juliana Alcoforado Diniz.

**Resources:** Dario Zamboni, Pegine B. Walrad, Petr Volf, David L. Sacks, Angela K. Cruz.

**Supervision:** David L. Sacks, Angela K. Cruz.

**Writing – original draft:** Juliana Alcoforado Diniz.

**Writing – review & editing:** Juliana Alcoforado Diniz, Mariana M. Chaves, Slavica Vaselek, Rafael Ricci-Azevedo, Renan V. H. de Carvalho, Tiago R. Ferreira, Pegine B. Walrad, Petr Volf, David L. Sacks, Angela K. Cruz.

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
