## [Decision Letter · Decision Letter 0]

22 Sep 2020

Dear Cruz,

Thank you very much for submitting your manuscript "Protein methyltransferase 7 deficiency in Leishmania major increases neutrophil associated pathology in murine model" for consideration at PLOS Neglected Tropical Diseases. As with all papers reviewed by the journal, your manuscript was reviewed by members of the editorial board and by several independent reviewers. In light of the reviews (below this email), we would like to invite the resubmission of a significantly-revised version that takes into account the reviewers' comments. 

We cannot make any decision about publication until we have seen the revised manuscript and your response to the reviewers' comments. Your revised manuscript is also likely to be sent to reviewers for further evaluation.

Sincerely,

Camila I. de Oliveira

Associate Editor

Ricardo Fujiwara

Deputy Editor

Reviewer's Responses to Questions

**Key Review Criteria Required for Acceptance?**

**Methods**

-Are the objectives of the study clearly articulated with a clear testable hypothesis stated?

-Is the study design appropriate to address the stated objectives?

-Is the population clearly described and appropriate for the hypothesis being tested?

-Is the sample size sufficient to ensure adequate power to address the hypothesis being tested?

-Were correct statistical analysis used to support conclusions?

-Are there concerns about ethical or regulatory requirements being met?

Reviewer #1: (No Response)

Reviewer #2: The objectives of the study are clearly articulated. The design is appropriate to test the hypothesis, some suggestions were made (additional coments for the authors). The sample size is sufficient to address th hypothesis tested and statistics used are aprropriate.

**Results**

-Does the analysis presented match the analysis plan?

-Are the results clearly and completely presented?

-Are the figures (Tables, Images) of sufficient quality for clarity?

Reviewer #1: (No Response)

Reviewer #2: The figures are of sufficient quality for clarity.

**Conclusions**

-Are the conclusions supported by the data presented?

-Are the limitations of analysis clearly described?

-Do the authors discuss how these data can be helpful to advance our understanding of the topic under study?

-Is public health relevance addressed?

Reviewer #1: (No Response)

Reviewer #2: The conclusions are supported by the data presented and the limitations of the study discussed. public health relevance is discussed.

**Editorial and Data Presentation Modifications?**

Reviewer #1: (No Response)

Reviewer #2: Minor comments:

1)Figure 2. Higher levels of PRMT7 mRNA are observed in L. major CC1, a non-pathogenic strain of L. major. Have the authors any data showing that there is a correlation between PRMT7 mRNA levels and the activity of the enzyme. This could lead to possible differences in PRMT7-dependent methylation in the different L. major strains with consequence on the PRMT7 gene control at post transcriptional levels. This should at least be discussed.

2) The CC1 L. major strain is attenuated however, it is not correct to state that “L. major CC1 did not cause any lesion” (p. 14 line 287). Figure 2C clearly shows the presence of a lesion which is smaller than those observed following infection with the other strains but clearly present. Please modify the text accordingly.

3) Although the authors state that they use the ear model of infection involving intreadermal infection with a low number of parasites (p. 19) they infect mice with a high dose of parasites (10e5). This should be corrected. 

4) Figure 2D ** is significant compared to what?

5) The authors conclude (p.22) that their data provide a novel link between neutrophil mediated inflammation and lesion formation. The correlation between neutrophil presence in unhealing lesions and lesion formation has already been reported by several groups (complete references). The novelty is the involvement of LmjPRMT7 in this process even if, whether neutrophils are the cause or the consequence remains to be determined.

6) PRMT7 is acting mostly post-transcriptionally thus it is not so surprising that only few differentially expressed transcripts were detected between LmjPEMT7KO parasites and WT strains. The authors have performed proteomics on this parasite stain, together with the present data this will lead to further studies. 

7) Typos in supplementary figure 1.

**Summary and General Comments**

Reviewer #1: Ferreira et al. report on their progress in characterizing the effect of protein methyltransferase 7 on parasite fitness and virulence throughout the life cycle in Leishmania major. This is a logic extension of their previous work but also a formidable task given that one expects a polyphenic phenotype in Lmprmt7 deficient parasites derived by genetic manipulation (as originally described in ref 13). 

The strengths of the work are related to the characterization of parasites’ phenotype in the major habitats encountered during the life cycle. In the sandfly vector quantitative development is severely affected and ‘metacyclic’ parasites morphs seem to be severely reduced if parasites are Lmprmt7 deficient. In contrast, lesion development in mice seems exacerbated and this correlates with an increased local presence of neutrophils. The experiments and approach to characterize these phenotypes are appropriate and well conducted. 

The major weakness in view of this referee is related to the interpretation of the results which is not straight forward because of inconsistencies with the previously published findings using the same parasites (ref. 13). A major issue is the apparent dramatic regain of pathogenicity with respect to lesion induction of the Lmprmt7 deficient strain which presumably is still the same as the one described in 2014. The previously published data (Fig. 4b in Ferreira et al. 2014, ref 13) suggest a very minor gain of pathogenic potential by deleting Lmprmt7. This has apparently changed dramatically since (Fig. 3A this work). It is noteworthy that, according to the authors, Lmprmt7 could not be deleted in LV39 another parasite isolate. This indicates on the one hand an essential role of the gene depending on genetic background, and, on the other hand, a strong selective pressure for compensating mechanisms to evolve to simply maintain viability. Thus, parasites originally described in 2014 may have accumulated compensatory mechanisms and this is not considered in the present work. In particular, should the strains have been passaged since several times in vivo which is a common routine in many labs working with L. major to maintain the virulent phenotype because that can be prone to lose it if only passaged in vitro. Thus, a simple approach to control for this is to re-complement the currently used Lmprmt7 deficient strain. 

A similar point of inconsistency over time is the behaviour of the L. major CC1 strains in vitro in macrophages where differences were claimed to emanate from Lmprmt7 deficiency (Fig. 6c this ms and Fig. 4a ref 13). 

Finally, Lmprmt7 may have moonlighting functions. Hence, complementation with a functional null mutant may also be worth considering. 

Minor points:

1. If indeed neutrophil response is functionally linked to Lmprmt7 presence then authors may contemplate using nude or rag-deficient mice that would eliminate the immunopathological component of the T cell response that is most prominent in BALB/c mice. 

2. The complemented strains are described as having ectopic copies of Lmprmt7 gene which is a sort of vague description given that episomally and genomically compensated strains were constructed originally. This should be specified as it further affects interpretation and review of the results. 

3. Quantitative real time PCR to compare Lmprmt7 mRNA levels is suboptimal given the mentioned reliance of gene expression on post-transcriptional mechanisms, protein and, more relevant, activity would be much more convincing. Moreover the choice of a pentose-phosphate pathway reference or house keeping gene can be misleading when comparing life cycle stages since the pathway is clearly regulated as well (e.g. Rosenzweig et al. 2007 for the situation in L. donovani which applies to other species too).

Reviewer #2: In this manuscript, Diniz et al report that one of Leishmania major protein methyltransferases, LmjPRMT7 has an impact on parasite virulence in vivo. High levels of LmjPRMT7 can impair the pathogenicity observed following infection of wild type mice. This was shown by restoration of virulence in an attenuated L. major strain upon LmjPRMT7 deletion. Although a significant impact was observed on lesion development following infection with L. major parasites deleted of LmjPRMT7, no impact was observed on parasite control 7 weeks after infection. In contrast, the absence of LmjPRMT7 impaired parasite development in the sand fly vector. Putative genes modulated by the absence of this enzyme are presented. The observations are clearly presented. 

Specific comments to the authors: 

Major comments: 

1) The authors conclude (p.23) that neutrophils rather than macrophages play a major role in the uptake of parasites lacking LmjPRMT7. However, 4 hours post infection, macrophages take up 35 % of parasites (with or without LmjPRMT7) while neutrophils uptake is of approximately 40% for LmjPRMT7-deleted parasites. Thus, the difference between macrophages and neutrophils does not appear to be a major one. a possibnle difference in parasite uptake by macrophages and neutrophils should be confirmed in vivo by flow cytometry analysis.

2) The main phenotype observed in mice infected with �PRMT7 parasites is the development of a necrotic lesion associated with the presence of neutrophils. The author hypothesize that sustained neutrophil recruitment to the site of infection is responsible for exacerbated disease (p. 21). To assess if increased migration of neutrophils is responsible for the bigger lesion observed, the authors should first investigate if increased neutrophil recruitment is observed in the first days (24 hours and 72 hours) post infection, two time points where neutrophils were shown to be recruited or not, respectively, to determine an impact of the lack of PRMT7 on neutrophil migration per se. Transwell experiments could also be performed to compare the neutrophil recruitment by the different strains

3) Figure 5: It is clear that the number of parasites per ear does not vary between WT, �LmjPRMT7 and the add back control. What is more surprising is that the number of parasites observed in the ears of C57BL/6 mice is quite elevated for mice of a resistant genetic background with values within the range observed in lesion of infected BALB/c mice. Could the authors comment on this.

4) The impaired formation of metacyclic �LmjPRMT7 in the sand fly should be discussed further

PLOS authors have the option to publish the peer review history of their article (what does this mean?). If published, this will include your full peer review and any attached files.

Reviewer #1: No

Reviewer #2: No
---

## [Decision Letter · Decision Letter 1]

28 Jan 2021

Dear Cruz,

Thank you very much for submitting your manuscript "Protein methyltransferase 7 deficiency in Leishmania major increases neutrophil associated pathology in murine model" for consideration at PLOS Neglected Tropical Diseases. As with all papers reviewed by the journal, your manuscript was reviewed by members of the editorial board and by several independent reviewers. The reviewers appreciated the attention to an important topic. Based on the reviews, we are likely to accept this manuscript for publication, providing that you modify the manuscript according to the review recommendations. 

Sincerely,

Camila I. de Oliveira

Associate Editor

Ricardo Fujiwara

Deputy Editor

Reviewer's Responses to Questions

**Key Review Criteria Required for Acceptance?**

**Methods**

-Are the objectives of the study clearly articulated with a clear testable hypothesis stated?

-Is the study design appropriate to address the stated objectives?

-Is the population clearly described and appropriate for the hypothesis being tested?

-Is the sample size sufficient to ensure adequate power to address the hypothesis being tested?

-Were correct statistical analysis used to support conclusions?

-Are there concerns about ethical or regulatory requirements being met?

Reviewer #1: (No Response)

Reviewer #2: The methodology is appropriate. 

The authors should specify if "control" mice are mice uninfected or mice injected with saline or PBS.

**Results**

-Does the analysis presented match the analysis plan?

-Are the results clearly and completely presented?

-Are the figures (Tables, Images) of sufficient quality for clarity?

Reviewer #1: (No Response)

Reviewer #2: The authors have answered many of the queries. 

However, there remains few points that should be better presented and discussed. 

In the Figure 6C the new data obtained for the ear macrophages are presented but not those for the ear neutrophils. These should be included, as indicated in corresponding Figure legend, even if they are not similar than waht is observed in vitro.

1. In the new Figure S4, the authors state that "there is no difference between the control and KO strains, since all L. major strains used show a low neutrophil recruitment rate in this early time points". The data clearly show a statistically significant difference in neutrophil recruitment in the ear dermis 24 hours p.i. between delta lmjprmt7 and the add back strain (denoted by stars) which show increased neutrophil recruitment. The increase observed between WT and deltalmjprmt7 does not appear significant in the experiment presented, the authors may indicate it this was the case in the other experiment performed. The authors should include in the "Discussion" that increased neutrophil recruitment at the onset of infection may have an impact on subsequent development of the disease, as reported by several groups, and may contribute in part to the pathology observed following infection with deltalmjprmt7 mice. Furthermore, while a difference in the frequency of cells was observed between deltalmjpmt7 and the add back strain, the difference in neutrophil number observed 7 weeks post infection (Fig.4) was not statistically significant between the ears of delta lmjprmt7 and the add back strain, suggesting that the differences observed in the cutaneous pathology may only partially correlate with neutrophil presence. This could be indicated in a few words.

The references cited (33-35) are not using L. major the authors should also include references including L. major that would be more relevant to this study.

**Conclusions**

-Are the conclusions supported by the data presented?

-Are the limitations of analysis clearly described?

-Do the authors discuss how these data can be helpful to advance our understanding of the topic under study?

-Is public health relevance addressed?

Reviewer #1: (No Response)

Reviewer #2: As mentioned above, the study is well presented and the results interesting. A few points should be included in the discussion, as suggested above.

**Editorial and Data Presentation Modifications?**

Reviewer #1: (No Response)

Reviewer #2: See comments under the Results section.

**Summary and General Comments**

Reviewer #1: (No Response)

Reviewer #2: The authors have answered most of the querries. Insertion of the the suggestions will complete the study.

PLOS authors have the option to publish the peer review history of their article (what does this mean?). If published, this will include your full peer review and any attached files.

Reviewer #1: No

Reviewer #2: No
---

## [Decision Letter · Decision Letter 2]

10 Feb 2021

Dear Cruz,

We are pleased to inform you that your manuscript 'Protein methyltransferase 7 deficiency in Leishmania major increases neutrophil associated pathology in murine model' has been provisionally accepted for publication in PLOS Neglected Tropical Diseases.

Best regards,

Camila I. de Oliveira

Associate Editor

Ricardo Fujiwara

Deputy Editor

Reviewer's Responses to Questions

**Key Review Criteria Required for Acceptance?**

**Methods**

-Are the objectives of the study clearly articulated with a clear testable hypothesis stated?

-Is the study design appropriate to address the stated objectives?

-Is the population clearly described and appropriate for the hypothesis being tested?

-Is the sample size sufficient to ensure adequate power to address the hypothesis being tested?

-Were correct statistical analysis used to support conclusions?

-Are there concerns about ethical or regulatory requirements being met?

Reviewer #2: (No Response)

**Results**

-Does the analysis presented match the analysis plan?

-Are the results clearly and completely presented?

-Are the figures (Tables, Images) of sufficient quality for clarity?

Reviewer #2: (No Response)

**Conclusions**

-Are the conclusions supported by the data presented?

-Are the limitations of analysis clearly described?

-Do the authors discuss how these data can be helpful to advance our understanding of the topic under study?

-Is public health relevance addressed?

Reviewer #2: (No Response)

**Editorial and Data Presentation Modifications?**

Reviewer #2: (No Response)

**Summary and General Comments**

Reviewer #2: (No Response)

PLOS authors have the option to publish the peer review history of their article (what does this mean?). If published, this will include your full peer review and any attached files.

Reviewer #2: No

---

## [Editor Report · Acceptance letter]

24 Feb 2021

Dear Cruz,

We are delighted to inform you that your manuscript, "Protein methyltransferase 7 deficiency in Leishmania major increases neutrophil associated pathology in murine model," has been formally accepted for publication in PLOS Neglected Tropical Diseases.

Best regards,

Shaden Kamhawi

co-Editor-in-Chief

Paul Brindley

co-Editor-in-Chief
